# Greenspace as Workplace: Benefits, Challenges and Essentialities in the Physical Environment

**DOI:** 10.3390/ijerph20176689

**Published:** 2023-08-31

**Authors:** Victoria Linn Lygum, Katia Dupret, Peter Bentsen, Dorthe Djernis, Sidse Grangaard, Yun Ladegaard, Charlotte Petersson Troije

**Affiliations:** 1Department of the Built Environment, The Faculty of Engineering and Science, Aalborg University, 2450 Copenhagen, Denmark; 2Research Centre for Social Entrepreneurship and Sustainability, Department of People and Technology, Roskilde University, 4000 Roskilde, Denmark; 3Center for Clinical Research and Prevention, Copenhagen University Hospital-Bispebjerg and Frederiksberg, 2000 Frederiksberg, Denmark; 4Department of Geoscience and Natural Resource Management, University of Copenhagen, 1958 Frederiksberg, Denmark; 5The Foundation for Mental Health, 2500 Valby, Denmark; 6Department of Psychology, University of Copenhagen, 1168 Copenhagen, Denmark; 7Department of Urban Studies, The Faculty of Culture and Society, Malmö University, 20506 Malmö, Sweden; 8Division of Sociology, School of Health, Care and Social Welfare, Mälardalen University, 72123 Västerås, Sweden

**Keywords:** outdoor office work, employee wellbeing, landscape architecture, green outdoor environments, nature, restorative environments, work life balance, workplace

## Abstract

There is a scarcity of knowledge regarding the potential benefits of human–nature contact within the context of working life. Even more limited is the research that focuses on working outdoors and the setting in which it takes place. This study aimed to obtain insight into key aspects of the physical environment relevant for the experienced benefits and challenges of workers exploring office work outdoors. We conducted interviews with key informants as well as photo registration and mapping of the different green spaces in the environments of six small or medium-sized workplaces. The information gathered was used as background knowledge for exploratory qualitative interviews, which were conducted while walking in natural settings chosen by the interviewees. With a landscape architectural perspective and an inductive approach, we explored employees’ experiences of bringing office work outdoors. The following themes emerged: ‘Simplicity,’ ‘Safeness’, ‘Comfort’, and ‘Contact with Nature’ were experienced as key aspects in relation to the physical environment, whereas ‘Sociality’, ‘Well-being’, and ‘Functioning’ stood out as the main benefits and, ‘Digital dependency’ and ‘Illegitimacy’ as challenges to overcome. Based on the identification of potential benefits and their prerequisites, we propose implications for practice and research that can be useful when focusing on bringing office work outdoors.

## 1. Introduction

### 1.1. The Benefits of Greenery

Nature places in urban outdoor environments play a vital role in the overall well-being and quality of life of people [1]. Research within a multitude of scientific disciplines points towards the importance and potential benefits of human–nature contact [2]. The body of knowledge about the restorative effects of greenspace is comprehensive, both regarding stress reduction and mental fatigue, the prevention of depression, and the enhancement of people’s mood and vitality [3,4,5]. Across various fields, there is a consensus regarding the value and impact of nature experiences on mental health, including emotional well-being and cognitive functioning [6]. In addition to these and other dimensions related to health and well-being, there are several other potentially beneficial perspectives to consider when investigating a greenspace as a workplace. Notably, there is a growing body of research on the potential of the outdoors to foster creativity [7] and facilitate learning [8]. Furthermore, there are several significant environmental factors, including natural sights, natural sounds and negative air ions. Exposure to these factors can contribute to various positive physiological and psychological states, such as relaxation, vitality and attention restoration. Exposure to the mentioned factors can also lead to positive health outcomes, including a decreased prevalence of cancer, cardiovascular disease and depression [2].

The positive association between exposure to natural environments and health and well-being is evident even with the rather crude measure of time, without taking the type of activity or various contextual conditions into account. In a study involving nearly 20,000 Britons who were asked about the time they spent in natural environments over the past seven days, the results showed that people were significantly more likely to report good health or well-being if they had spent more than 120 min in natural environments. The positive associations peaked at 200–300 min per week, and this was true regardless of whether the time was spent in many short or few longer periods [9]. In another study, a scoping review of college-aged adults’ time spent sitting or walking in nature revealed beneficial effects on their mental health after as little as 10–20 min [10].

The possible pathways are numerous and, so far, not fully understood [11,12]. The objective of the subsequent paragraphs is to assess the existing body of research, with a specific emphasis on identifying the key variables that influence the implementation of outdoor office work.

### 1.2. The Benefits of Greenery to Working Life 

Despite much evidence of positive relations between exposure to natural elements and human well-being and health in general [2], as well as the more specific topics of mental health [6], stress reduction [13,14], attention restoration [15], cognitive performance [16], creativity [7], and learning [8], research relating this knowledge to the area of working life has been scarce until recently. However, research conducted in the context of the workplace has shown that greenery can contribute to lower levels of stress, as well as to a more positive attitude towards the workplace [17]. When addressed, it is often from the point of view of the indoor environment—and its connection to the outdoors. Several studies have shown that a window view of natural elements or environments (e.g., trees, lawns, shrubs, or flowering plants) at the workplace can have a positive impact on the well-being of employees [18,19,20]. Moreover, it can affect job satisfaction positively [21,22,23,24] and lower employees’ stress levels [20,25,26]. Consistent with this, Lottrup et al. [27] found that a view of a green outdoor environment from inside the workplace was associated with high workability.

Moreover, if employees take breaks in a green outdoor environment, their stress levels are significantly lower than if they stay indoors [28]. Such breaks may also contribute to improving the employee’s quality of life in general [29]. However, there may be significant advantages to not only having a “green window” or taking outdoor breaks, but also moving work outside. Research studies indicate that exposure to natural elements and environments can lead to increased attention and improved cognitive performance in solving various tasks [30,31,32,33,34], as well as enhanced alertness [26]. The use of green space for tasks can furthermore have a positive effect on creativity [35] and can impact employees’ enthusiasm and satisfaction with work [36]. In summary, there are a multitude of positive effects associated with bringing work out into green surroundings. Several theories have been proposed to elucidate these findings, with the most prominent theories being elaborated below.

### 1.3. Theories Exploring Restorativeness and Preferences in Regards to Natural Environments

Among theories explaining the benefits of being in contact with natural environments, one of the most influential is the Attention Restoration Theory [37]. It suggests that some environments require attentional effort which can lead to mental fatigue while others allow attentional rest and restoration. The theory further suggests that the restorative environments have four essential characteristics that are likely to be found in natural settings. These are ‘fascination’, ‘being away’, ‘extent’ and ‘compatibility’. Fascination is what happens when experiencing a sunset or the movement of leaves in a slight breeze, stimuli that calls for involuntary attention, and which allows both reflection and recovery. Being away implies that you leave the activities that require attentional effort and that you find escape or withdrawal from mental effort and nuisance. Extent refers to the sense of being in a completely different world, a setting that gives the experience of connectedness and scope. Finally, compatibility is where one’s purpose fits the demands of the setting and where you find the facilities and information needed for action.

Another theory relevant to understanding preferences for certain features of natural environments is the Prospect–Refuge Theory [38]. It claims that humans prefer landscapes that afford good scope for both refuge and prospect compared to landscapes that have neither. Refuge is about the opportunity to hide, and prospect implies an unhindered opportunity to see. The preference is linked to an apparent potential in landscapes to satisfy many of our biological needs, such as food, rest, and escaping danger [38]. Both the four characteristics that are important for a restorative experience and the preference for landscapes that afford both prospect and refuge can be useful when studying preferences for natural environments and, in this context, outdoor settings for office work.

### 1.4. The Significance of the Outdoor Setting

Although research studies have addressed the benefits of exposure to natural elements and environments at workplaces [18,20,27,28,30], there is only a little focus on issues related to working outdoors and which settings the workers prefer. Access to green space, such as a roof terrace or a park in the city, has until recently been scarce in the leading architectural literature about office spaces [39,40,41,42]. The focus in office spaces is primarily on the creation of different kinds of spaces and facilities supporting the activity of the office, such as spaces for brainstorming, lounges and quiet rooms. Grangaard [43] criticized this literature for the lack of focus on the connections and shifts between activities and spaces. However, in a recent literature study, Hu et al. [44] focused upon knowledge workers and the relation between landscape preference and health benefits from a theoretical perspective. They proposed four theoretical mechanisms which may promote the health of knowledge workers: stress reduction, attention restoration and landscape preference, physical activities promotion and sensory enrichment. In addition, they identified 29 key characteristics of restorative environments that can promote knowledge workers’ well-being and health, for example ‘being away’, ‘complexity’, ‘safety’, ‘deflected vistas’, ‘legibility’ and ‘compatibility’. Hu et al. [44] pointed to the importance of further interdisciplinary research, including empirical studies such as interviews and on-site observations.

In the interactive research project ‘StickUt Malmö’, the potential benefits and challenges of bringing work, traditionally conducted indoors, outdoors, were explored [45]. The findings showed that many different work activities could be taken outdoors in a variety of ways, both individually and in collaboration with others. The participants (58 civil servants from the City of Malmö) associated their outdoor work with many positive experiences, such as a sense of well-being, enhanced cognition, recovery from stress, autonomy and better communication and social relations. Feelings of guilt and illegitimacy were, however, also prevalent. Apart from the importance of supporting workplace culture and leadership, it was evident that easy access to greenspaces and pedestrian paths were crucial for outdoor office work to occur and function well [46].

The studies cited in this paper have provided valuable insights into the positive effects of spending time in natural environments on mental health, job satisfaction and cognitive performance. However, they have not fully explored the specific requirements and preferences for outdoor settings in the context of office work. To bridge this gap and to advance knowledge in the field of outdoor office work, the project Pop out! was developed. Inspired by the Stick Ut project [47], the Pop out! project seeks to provide a more comprehensive understanding of the prerequisites and preferences related to outdoor office work. We explored employees’ experiences of taking work outdoors in different types of greenspaces, investigating the challenges of going outside as well as the relationships between employees’ work tasks, outdoor settings and experiences. Understanding the relationships and experiences will help identify key factors for successful implementation and inform landscape architectural interventions. This knowledge is important for creating work environments that may promote well-being, enhance productivity and offer a positive alternative to traditional indoor office spaces. Thus, the purpose of this study was to gain further insight into key aspects of the physical environment relevant for the experienced benefits and challenges of workers exploring office work outdoors.

## 2. Materials and Methods

### 2.1. Case Selection

Pop out! is an organisational intervention project aiming to systematically support the changes in attitude and habits that are required to bring office work tasks outdoors. The applied research-based process tools are evaluated in a mixed-method counterpart study. 

The study included five private small and medium sized companies that were reached through social media or contacted directly. The sample selection criteria for the inclusion of companies were pragmatic and primarily based on the individual organisation’s motivation for participation, a geographical diversification among the participating companies (to strengthen geographical representativeness) and an assessment of the individual company’s scope to include a relevant outdoor environment that both created variety between the participating companies and ensured varied experiences. Employees were invited to attend a process of identifying tasks from their indoor office work and bringing them outdoors. For two months, different types of outdoor activities related to work were facilitated by process consultants with specific knowledge in mental health, organisational change, and outdoor office work. Three 45 min meetings were held: A kick off meeting, a webinar and a workshop at the end of the intervention. Inspiration and learning videos were produced and made accessible to the participants (Figure 1).

To gain further insights into key aspects of the physical environment, companies located in different types of greenspaces were included. The case selection was thus information-oriented with a sample of cases, each with their different qualities [48] and focused on including a rich variety of greenspace, where the term ‘greenspace’ is defined loosely to include any natural space such as terraces, courtyards, gardens, parks, streetscape greenery, harbour areas, beaches, bush lands and forests.

One of the companies had two different locations, resulting in a total of six cases (see Table 1), each with an average of 17 voluntary participants. Among the participants, the management selected eight interviewees, who were deemed to be especially informative due to their high level of participation in the project. All the selected interviewees were chosen for their ability to provide detailed accounts of the challenges and advantages associated with working outside. In this regard, the selected interviewees can be considered to be the best sources for acquiring the most qualitatively rich and detailed accounts and experiences [48]. The selection of interviewees, based on their level of informativeness and participation, led to a sample consisting of eight female interviewees. Their age ranged from 22 to 61 and all of them were office workers with a job function that required access to a computer. The Email marketing specialist, technical designer, customer service employee, manager of a working environment and chief accountant were mostly bound by computer work. Whereas the consultant, sustainability manager and head of lettings undertook both computer work and more active work tasks such as showing customers around. The term ‘case’ is employed to refer to each of the six locations, with one or two interviews conducted per case.

### 2.2. Data Collection and Analysis—Research Phase One

The first phase of the study involved conducting unstructured individual interviews with key informants at each of the six cases centred on the topic of outdoor office work. The interviews lasted approximately one hour and were held at the company facilities, starting indoors and ending with a walk in their outdoor environment. Questions concerned how they worked indoors, what options they had for working outdoors and how they used those options. The interviews were also about positive and negative aspects of working outdoors, as well as their preferences and wishes in this regard. Furthermore, they featured a photo registration and mapping of different types of outdoor settings in their surroundings shown by the interviewees on the walk during the interview. Subsequently, a description was compiled based on the collected data about spatial characteristics, amenities and potential activities available within the outdoor settings.

Apart from serving research purposes, the descriptions were used at the kick-off to introduce the participants to the possibilities in their outdoor environments for office work.

### 2.3. Data Collection and Analysis—Research Phase Two 

Phase two was carried out right after the exploration period and included individual walk-and-talk interviews with the selected interviewees in each case. The interviewer was the same person who had carried out the research of the first phase of this study–a researcher within the field of landscape architecture with expertise in connections between human health, nature, and design. The interviews were carried out while walking through the different greenspaces, which the interviewees had used during the active testing. The walk-and-talk also included a photo registration of the outdoor settings discussed during the walk. The eight interviews lasted approximately 30 min each and were semi-structured [49], covering several topics of conversation. The topics revolved around their use of greenspace for work, positive and negative aspects, needs and preferences for settings and activities, as well as benefits and challenges of working outdoors. The topics were designed to capture the nuances and breadth of experiences in conducting work outside. Conducting the interview in the outdoor setting gave the interviewer a good understanding of the connections between the experiences discussed and their relation to specific aspects of the physical environment. The photo registration helped capture these connections. The seven stages of an interview inquiry described by Kvale and Brinkmann [50] assisted in the planning of the interview study. The recordings from the interviews were transcribed and analysed. Thematic analysis [51] was used to analyse the transcriptions in connection with the data from the photo registration and phase one. The focus was on relating the interviewees’ statements to the specific greenspaces they talked about, bringing into play the information gained through the registration of their outdoor environment. The thematic analysis was driven by the search for issues related to going outside, as well as the relations between employees’ work tasks, outdoor settings, and experiences. The steps of the analysis were as follows: getting to know the data, generating initial codes, identifying themes, reviewing themes, naming themes, and writing up themes [51].

## 3. Results

This section unveils the findings from the data collection and the thematic analysis of interviews conducted across six different cases. Each case represents a unique context, company, and outdoor space, offering insights into the benefits and challenges associated with working outdoors and dimensions of importance in relation to the physical environment. The findings, presented in eight themes, offer guidance on the design and considerations necessary for creating successful outdoor workspaces.

### 3.1. The Six Cases 

Case 1 is located in the periphery of the city in one of Copenhagen’s newest districts, close to a large nature park consisting of wetlands, young forests, and a system of canals and lakes (see Figure 2a). It is home to a variety of animal life, including birds and deer. The company, an e-marketing agency, resides in a block of flats and offices typical for the district. It is situated on the second floor, and the office space is mostly open-plan. The nearest outdoor environment is a landscaped courtyard with a few seating opportunities. In the street, green boulevards, an outdoor fitness/play area, and a water reservoir connect the district to the nature park. The interviewee from case 1 is an email marketing specialist who is dependent on her computer for most of her work tasks.

Case 2 is situated in a rural area surrounded by fields as well as spruce and beech forest (see Figure 2b). The company, a contracting company specialised in lifts and balconies, is housed in a one-story office building with two terraces and a lawn accessible from the open-plan office. These immediate surroundings afford a view to a cow corral and the forest. Interviewee 2 is a technical designer working primarily with architectural drawings on her computer.

Case 3 is located in the centre of Copenhagen on the harbour front (see Figure 2c). The company is a trade association within the field of the built environment and is part of an office community that has access to several roof terraces, an urban square and a landscaped courtyard. The near surroundings feature a waterfront promenade, sidewalks along canals and an urban park. Interviewee 3 works as a consultant in charge of planning and executing events and courses. Her work tasks are both computer-based and more active.

Case 4 is part of a park-like company campus located in a suburban area (see Figure 2d). The company resides in a multi-story office building with different types of offices. From the building there is access to an entrance area with several seating opportunities as well as a terrace furnished with tables and chairs. A path system leads through the wooded campus, giving access to courtyards, gardens, and views of ponds and lakes. The campus area offers various types of seating, such as benches and picnic tables. Interviewee 4 is a Sustainability Manager with both computer-based and more active work tasks. 

Case 5 is located in a coastal country town (see Figure 2e). The company is situated in a one-story office building that mainly consists of multi-person offices. From the building there is access to a terrace, a garden and the adjacent beach and small harbor. The surroundings also feature a cow corral, detached house quarters and a park. Interviewee 5a is a manager of the working environment and interviewee 5b is a chief accountant. Both primarily have work tasks dependent on a computer.

Case 6 is part of a suburban campus for companies and a university (see Figure 2f). The company resides in a multi-story office building with different types of offices. It provides direct access to terraces with tables and chairs. In the near surroundings, the campus includes courtyards, gardens and squares, as well as streetscape greenery. Furthermore, a path system leads through wooded areas surrounding the campus. Interviewee 6a works with customer service, and many of her work tasks are connected to the reception, whereas interviewee 6b works with showings and has both computer-based tasks and showings.

### 3.2. Themes

The following themes emerged through the thematic analysis, highlighting various benefits and challenges with regards to bringing office work outdoors. Additionally, the themes help identify the significant features and types of settings necessary for successfully establishing and maintaining outdoor office work environments.

#### 3.2.1. Simplicity

In one way or another, interviewees talked about the importance of it being easy to take work outside (1, 2, 3, 4, 5a, 5b, 6a). To provide an example, Interviewee 3 emphasised the need for simplicity and minimal planning in relation to working outdoors. Interviewees mentioned that the successful solutions included those that were quick and easy, such as sitting out on a terrace to read, eat lunch outside, or take a mobile phone call outdoors (1, 2, 3, 4, 5a, 5b, 6a) (see Figure 3). Among other things, it was mentioned that it made things a lot easier if the outdoor area was close by: “When the phone rings, I put on my headset and then I just open the door and walk straight out onto the terrace—so it’s easy!” (5a). In this context, Interviewee 5a pointed out that a paved surface was a deciding factor for her because then she could go outside irrespective of which footwear she was wearing. 

Interviewees also pointed out other things that facilitated going outdoors. For instance, for it to be part of their routine (1, 5b and 6a): “I think it is about getting a habit…a routine…you know that when you have been outside taking a break, you’re ready for the remaining four hours, right…” (1). Being able to easily adapt or choose their route in relation to the time needed and the task to be solved was also mentioned as an important factor for successful outdoor work (1, 5a, 6a): …”and the awareness that you can walk further and adjust the length according to your talk… because when you come back to the office, the conversation that you had outside always ends promptly”. (5a)

In summary, the interviewees talked about simplicity, in the sense that working outdoors should not demand too much planning or investigation of the outdoor environment. They preferred an accessible greenspace close by, a fixed ‘going outdoors-routine’ and information/knowledge about routes with a flexibility in length as factors that could promote working outside. 

#### 3.2.2. Safeness 

Interviewees described how they wanted to avoid having to watch out for and be bothered by cars and bicycles (1, 5b, 6a). In the middle of the walk, interviewee 5b suddenly said: “let’s go this way…it is much calmer here where we do not need to watch out for cars”. Interviewee 6a specified that her favorite greenspace was a place where cars do not drive back and forth. In line with this, Interviewee 1 mentioned calmness and a minimum of traffic as assets of the route she liked and took daily together with a group of colleagues at lunchtime. Part of this route goes through a nature park and offers views to water, woodland and a common area.

Interviewee 3 said that it was important for her that the space she was in felt cozy and secure. In relation to this, she depicted an urban park featuring benches nestled amidst greenery and strategically shielded from the wind. Similarly, Interviewee 2 described her preferred location as follows: “I like to have a view but also to have a forest around me (see Figure 4). And then to sit in a sheltered spot. I like to be able to withdraw, but for it to still be open in the other direction. I guess I like that it feels like a cave.”

To recapitulate, interviewees indicated a need for feeling safe and secure in relation to the outdoor environment. This theme is related to how disturbing elements, such as traffic and wind, can create feelings of insecurity and hinder relaxation or the focus on work. On the other hand, the theme is also concerned with how certain qualities such as shelter, views and calmness can promote feelings of safeness.

#### 3.2.3. Comfort

Comfort was one topic that interviewees touched upon in different ways (see Figure 5). Rain and cold were mentioned as hindrances (1, 2, 3, 4, 5a, 5b, 6a, 6b): “I have been outdoors a lot…but only in good weather. I haven’t been out in bad weather…I just haven’t…it’s simply too inconvenient…it gets weird… simple as that” (4). Direct sunlight was also mentioned as an obstacle (3, 5b): “It means everything that we are using the terrace…. Before, we sat in front of the building where there is one hundred percent sun. That’s tough when you’re a redhead. Now, we both have sun and shade available.” (5b).

In the context of talking about the weather, interviewees expressed a wish for an orangery-like building (3, 4, 5b, 6a, 6b): “Well, I think we have really good conditions in spring and summer. We’ve sometimes talked about perhaps making an orangery that could be used in the winter–for meetings, for example. Maybe with some heating” (5b). In describing the building, the interviewees emphasized sitting comfortably in a cozy place in a bright and green environment and being sheltered. 

In addition to the weather, the risk of getting dirty was also mentioned as a challenge (2, 4, 6b): “Well, something that I’ve been thinking about is clothes getting dirty. It sounds very basic, but if you have an outdoor meeting at 9 a.m. and you get huge bird droppings on your clothes, it’s not very pleasant if you have to meet clients later in the day” (4). However, interviewees also talked about how to overcome these challenges by switching to appropriate footwear, remembering seat pads and having the option to bring an umbrella on a rainy day (2, 4, 5b).

To sum up, the weather played a major role in whether work was taken outdoors. Rain, cold and direct sunlight were mentioned as decisive factors. In the same way, getting one’s clothes dirty was a barrier for using the outdoors. However, interviewees also gave solutions to these challenges, listing things like appropriate wear and equipment. Moreover, the description of a sheltered space recurred in several interviews when asked about an ideal space for working outside. 

#### 3.2.4. Contact with Nature

Interviewees described getting away from noise and other distractions indoors to a place outside where they found peace (1, 2, 3, 4, 5a, 5b, 6a, 6b): “With music and chatting and phones and. So, it’s nice to get a bit of peace and quiet now and again” (1). In this context, interviewees also said that they avoided noisy places (1, 3, 6a): “But I don’t use this waterfront–it’s far too noisy, so I can’t focus properly enough to do anything” (3).

The possibility of going out to a place with unique sensory impressions was also a topic mentioned. The impressions emphasized varied from interviewee to interviewee and included fresh air, beautiful surroundings, scents, bird sounds and wildlife (1, 2, 3, 4, 6a, 6b): “I sat on one of those cushions you use in the garden, and the way you could kind of hear the sound of birds singing from out there, instead of sitting in an office and being interrupted” (6a). “I’d just like to show you a place that’s incredibly cool. A really odd place to go and have as a favorite spot too. But you’ll see. Someone has set up a feeder here, so there are just so many strange birds. The building is no architectural gem, but the experience is something, with all the life there” (4).

Whole landscapes were also described (2, 3, 4, 6b): “The building is beautiful and there are always flowers. When the trees bloomed and there were chestnuts, it was really beautiful. It’s such a holistic experience to sit here. It’s a really lovely place to sit” (3). “Walk into the woods and find a tree stump or something–there are lots of options. We’re really lucky in that respect–that we don’t just have to go to a park. That it is not set up. It’s nature-nature–not just a lawn” (2).

Furthermore, some liked the views, openness and brightness (1, 2, 5b): “Yes, I like the water. And the wide green expanses, they also contribute with something” (1). “It’s probably this, I think, that elevates this route above the others. You have air and spaciousness” (5b) (see Figure 6). While others talked about more enclosed spaces such as forests and parks (3, 5a): “Well for me it’s a forest a forest just has something in terms of colour and tranquillity” (5a). “I’m not so attracted to the boundlessly open. I hadn’t thought about it before, but everyone is different” (3).

Overall, taking work outside allowed interviewees to have a peaceful break from their office environment. Interviewees mentioned peace and different kinds of experiences of nature as qualities of the outdoor environment. Sensory impressions as well as varied types of whole landscapes were described as assets that were sought and appreciated.

#### 3.2.5. Sociality

The interviewees provided different examples of relations and cooperation with colleagues outdoors (1, 2, 4, 5a, 5b). Interviewee 1 mentioned the daily walk just after lunch as a good opportunity to chat with others: “So, it’s also very nice to have a chat, you know? That not everything is work-related.” Interviewee 5a noted that conversations outside naturally took on a less formal character compared to those indoors: “Yes. What also happens when you go outside is that the talk becomes less bound by agendas. Suddenly, you see something, and you talk about it. One guy on my team is really into butterflies, so even if we were talking about something work-related, he’d just say “have you seen”, and then our chat would be about butterflies.” Interviewee 4 focused on the fact that being outdoors could benefit creative collaboration: “I can’t say we wouldn’t have got there as quickly if we were sitting in the dark in a meeting room, but it just seemed to glide by relatively easily for us.” According to the interviewee, the collaboration process thus felt more smooth and effortless outdoors.

There were also interviewees who talked about walk-and-talk meetings (2, 5a, 5b): “For me, I probably feel there’s less pressure when you walk here. It’s more informal than those boss-employee roles where one asks questions and the other answers… Previously, I was a bit like ‘oh no, are we going to have a one-on-one again’… it’s helped me a lot to look forward to being able to meet when we’re walking around anyway … they’re more productive than the one-on-one meetings we have. I’m also more relaxed” (5b). Additionally, the interviewee knew that she was at a safe distance from the eyes and ears of others. 

Walking next to each other and sometimes behind each other seemed less confrontational than sitting across from each other in a meeting room (see Figure 7). And the time was fixed in a different way when both parties knew how long it would take to walk the route (5b): “Yes, and you know when we’re not back yet, that there are maybe 20 min left. So, you might as well just bring it up, if you don’t think something is going well or if there was a conflict the other day” (5b). The fixed time could also motivate people to get started on and resolve a task. In response to this, Interviewee 4 observed that the chilly spring weather had naturally put a limit on how long they could sit outside.

One aspect mentioned by Interviewee 3 was that working outside had, in a way, given the workplace an expanded inclusivity in that it was possible for an individual employee to withdraw and get some air outdoors if the need arose: “…the freedom it gives to be able to go outside…I think that’s quite important because we are all different…it gives space to withdraw. To get away” (3).

Summing up, the interviewees gave different examples of how working outside the office can be beneficial for relations and cooperation with colleagues. It made informal chatting and one-on-one meetings easier, enhanced creativity and aided in task solving. Furthermore, it was mentioned that having the opportunity to go outside made the work environment more inclusive, allowing individuals to take a break from or avoid an uncomfortable indoor situation. 

#### 3.2.6. Well-Being and Functioning

Interviewees gave examples of how going outdoors affected them (1, 2, 3, 4, 5a, 5b, 6a, 6b). Among other things, they said that they wound down and became calm in a different way than they were indoors (1, 2, 5a, 5b, 6b). Interviewees also mentioned that their mood improved (2, 4, 5a). One interviewee, 5a, said the following about walking in the woods in her free time: “You can’t walk in the woods and be angry! You just can’t. I’ve experienced being angry when I go out, but I always come back happy.” Interviewee 4 had the same experiences in terms of mood, which also influenced her job satisfaction: “Somehow it makes you happy. And when you’re happy and have that chemistry inside your body, I think you just get new ideas more easily, and things get easier. Things aren’t as heavy.” Interviewees also experienced a shift in creativity (1, 2, 4, 5a). Interviewee 1 had only experienced this when she had time off work and pointed out that for her it could not be forced but came quite naturally when her thoughts were just allowed to flow, for example, cycling to and from work or on a run. In addition, an enhanced ability to concentrate was also mentioned as a beneficial effect (6b), as well as the pleasure of exercise (1, 2, 4, 6b): “Especially when you’re just sitting on your behind all day—it just gives you something. In fact, it’s usually after lunch that we walk” (2). “It’s great if you can combine getting exercise with doing your job and getting some fresh air. Whether you’re having discussions while walking around, or phone conversations, or you’re sitting out having lunch, you know?” (6b) (see Figure 8).

To summarise, the interviewees gave different examples of how going outdoors had a positive impact on them. Several aspects were mentioned including calming down, improved mood, enhanced creativity, better ability to concentrate and experiencing the pleasure of getting physical activity.

#### 3.2.7. Digital Dependency

Interviewees also discussed hindrances. Interviewee 5b was restricted in getting outside because her phone was a landline and she sat in the reception. Dependence on one or more computer screens to perform one’s job was also mentioned as a key obstacle to getting outdoors (1, 2, 3). There were also divided opinions on whether sunshades for computers work or not: “We bought these leather sunshades… It works but you must look through this tunnel and they are quite difficult to fold out. It can be challenging if you are not the patient type” (3). “I waited some time to get my computer shade…then I went outside to work and in fact I sat there for a whole hour…that was a great experience (6a)”.

#### 3.2.8. Illegitimacy

In different ways, the interviewees felt ambiguous about bringing their work outdoors during the workday. For example, Interviewee 4 said: “I like the route behind the lake house it’s nice and there’s a special atmosphere over there you feel very close to nature. You almost feel that you ought not to be there during working hours.” Interviewee 3 had a similar expression of ambivalence: “There is probably something in all of us, even though we know very well that it is legal, and that the management wants it and that everything is good. Then there is still something in you. Either way, it’s there”. In line with these expressions, Interviewee 6a pointed out a need for reassurance: “It’s this thing… that it is legal to do it… without any of your colleagues thinking ‘oh now she is outside wandering around again’… I think we need to articulate its legality occasionally so that we do not fall back into old habits”. Interviewee 6b expressed the same need: “This culture where it is acceptable… you need to acquaint yourself with it…I mean that it is something that is actually allowed”. Interviewee 4 also touched upon the challenge when she described how some staff teased those who took their work outdoors. She said that they did this because they thought the project was a bit silly. On the other hand, Interviewee 6b mentioned that the COVID-19 crisis had helped to expand the boundaries of what one can and cannot do in terms of where and when we work.

To sum up, the interviewees’ expressions relate to a sense of illegitimacy when working outdoors. Although the project had been approved and was supported by management, the interviewees still expressed difficulty getting used to the fact that they were permitted to go outside. This indicates that it can be difficult, on an individual level as well as an organisational level, to change rooted ideas about what is appropriate and inappropriate in a work situation.

## 4. Discussion

### 4.1. Results Related to the Research Question, Previous Studies and Theories

In conclusion, the presented themes highlight aspects of the physical environment that are relevant to office workers’ experience of benefiting from doing office work outdoors. To briefly restate, the theme of ‘Simplicity’ refers to the ease of access to the outdoor environment, with convenient proximity to the office and established routines for going outdoors. ‘Safeness’ entails the absence of disruptive elements like traffic and wind. ‘Comfort’ involves the suitability of the outdoor environment for work, with a preference for favorable weather conditions or sheltered areas. ‘Contact with Nature’ highlights the appreciation for outdoor work experiences that encompass diverse encounters with nature. The theme of ‘Sociality’ suggests that office workers may experience enhanced collaboration and cooperation with colleagues and a more creative, informal, and inclusive work environment when the aforementioned aspects of the physical environmental are present. ‘Well-being and functioning’ capture the advantages of working outdoors in terms of improved mood, heightened creativity, and enhanced concentration, among other positive factors. Challenges mentioned by workers include the difficulty of performing tasks outdoors, particularly when dependent on computer screens, reflecting ‘Digital dependency’. ‘Illegitimacy’ arises when the organisational level does not explicitly permit or support the practice of outdoor work.

The finding of this study therefore reveals several perceived benefits, including improved relationships and cooperation with colleagues, reduced stress levels, enhanced mood, increased creativity and concentration, and the opportunity for physical exercise. Several of these findings align with earlier studies that indicate that the sight and use of natural elements and environments can lower stress levels [28] and improve attention, task solving [30,31,32,33,34], alertness [26], as well as creativity [35].

There are some obvious parallels between some of the themes generated in our study and the characteristics of restorative environments, as proposed in the Attention Restoration Theory [37]. The concept of ‘Compatibility’ may be related to Simplicity and Comfort, as it implies that the environment should align with the desired activities and encompass the need for an easy and meaningful transition from office work to an outdoor setting. Similarly, the idea of ‘Being away’ can be linked to the theme of Contact with Nature, as it involves disengaging from effortful activities and environments. This is reflected in the interviewees’ desire for a peaceful break from their busy and noisy office environment. With this said, there is a contradiction in the fact that the interviewees express a need to be away while at the same time taking part in what is effortful with them as they go outside. It may be argued that the interviewees exchange the tiresome aspects of the indoor work environment with a less formal and restorative setting and, in this exchange, still experience being away to some degree. The concepts of ‘Fascination’ and ‘Extent’ also intersect with the theme of Contact with Nature, where Fascination is seen in the focus on natural stimuli, and interviewees, who valued different qualities of outdoor spaces, expressed a focus on natural environments and sensory impressions. ‘Extent’ refers to an environment that offers rich experiences capable of capturing the mind, which aligns with how the interviewees described their preferred greenspace as a comprehensive and immersive encounter with nature.

The theme of Safeness encompasses the need for a greenspace that provides a sense of security, like a cave with a view of open space. This aligns with the Prospect–Refuge Theory [38], which suggests that humans prefer landscapes that offer both refuge and prospect.

When comparing the findings to the theoretical study conducted by Hu et al. [44] on knowledge workers, landscape preference, and health benefits, several similarities emerge. These similarities encompass mechanisms and characteristics such as safety, pathways, biodiversity, and compatible facilities.

Additionally, numerous parallels can be drawn to the results of the ‘StickUt Malmö’ project [45,46]. Both projects demonstrate that various work activities can be conducted outdoors, including one-on-one, walk-and-talk meetings, collaborative work with colleagues and individual tasks. Moreover, both projects highlight similar benefits, such as enhanced well-being, improved work ability, better communication and social relations. The issue of illegitimacy was also identified in both projects. In terms of the physical environment, both projects emphasise the importance of easy access to greenspaces, appropriate furniture, sun-/rain-/windshields and pedestrian paths as facilitating factors for outdoor office work. The project ‘Pop out!’ contributes with more detailed findings on different types of greenspaces and their features, revealing individual preferences for open or more enclosed spaces, the influence of sensory qualities such as sounds and scents and the connections between features in the environment and feelings of safeness, simplicity and comfort. These features include opportunities for prospect and refuge, minimal disturbance from traffic and availability of suitable equipment and attire, as well as paved and/or sheltered adjacent areas.

### 4.2. Limitations of the Study

The significance of the findings in this study should be acknowledged in relation to the case selection strategy, which deliberately focused on interviewees who were excep-tionally informative and highly engaged. The companies were selected for their qualities with regards to greenspace, covering different types of natural elements and environments. Furthermore, the interviewees picked out were those that, among the participants, had the most experience working outside during the test period. Two of the interviewees were anchor-persons for project ‘Pop out!’, which added to their dedication to the project. In this study, the participants who were most committed to working outdoors and therefore selected as cases, were female. As a result, the gender composition of the interviewees may have impacted the findings related to the benefits and challenges associated with working outdoors. If the study had focused on especially problematic cases [48], both with regards to the setting and interviewees, results would probably have been different accordingly. In fact, a good share of employees at the companies chose not to participate in the study. This de-selection may be due to different factors such as personal preference for using the outdoors or an incompatible job function with regards to bringing work activities outdoors. Moreover, the test period was during a favorable time of year for being outdoors. Had it been in the two winter months, findings would most likely have been different. The purpose of employing an information-oriented case selection strategy, specifically targeting particularly good cases [48], was to maximize the amount of information obtained from a small sample size. The more outdoor experiences, with the more varied greenspace, the more varied information concerning working outdoors. The topic of this study is in its in-fancy and needs to be further explored. Transferability may be limited due to various con-textual aspects such as type of business, employee characteristics, organizational culture, outdoor environment and climate [48,52]. Still, the findings of this study may serve as a useful building block for enterprises aiming at integrating the outdoors into an everyday working life.

## 5. Conclusions

### 5.1. Implications for Future Research

Research exploring the potential benefits of human–nature contact in the context of working life is still limited. Even more limited is the research that focuses on issues related to bringing office work outdoors and preferences for such settings. This study points to an array of both psychological and landscape architectural aspects to be further explored. This includes investigating the outcome of outdoor working in relation to creativity and health, the notion of the workspace in relation to shifts in activity and change of place, and challenges such as experiences of illegitimacy and barriers related to the outdoor environments. However, the study also suggests looking more into facilitators such as preferred settings, features and equipment, with a goal of identifying health design guidelines for supportive work environments that can maximize the benefits of working outdoors. These would be relevant avenues of exploration in order to study the ideal design of an outdoor adjacent construction specialized for outdoor work. It could also be relevant to further explore the design of settings that can foster different types of work, such as creative collaboration or individual learning, outdoors.

### 5.2. Implications for Practice

The aim of this study was to gain insights into the benefits and challenges of working in various types of greenspaces and to identify factors of key importance in the physical environment to support outdoor office work. In order to overcome the challenges framed by the themes Digital Dependency and Illegitimacy, and to capture the potential benefits of the themes Sociality, Well-being, and Functioning, there are some aspects of key importance in the physical environment. The identified themes of Simplicity, Comfort, Safeness, and Contact with Nature can be seen as aspects promoting outdoor office work—for it to come about and be rewarding. These are all key factors in facilitating the process of bringing office work outdoors. In Table 2, we make some suggestions and give examples of how these may be turned into practice.

## Figures and Tables

**Figure 1 ijerph-20-06689-f001:**
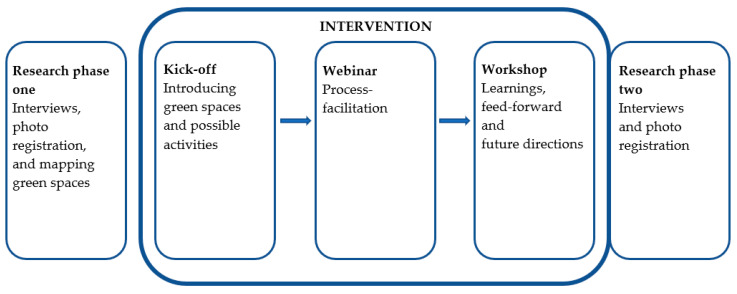
Phases of the Pop Out! Project.

**Figure 2 ijerph-20-06689-f002:**
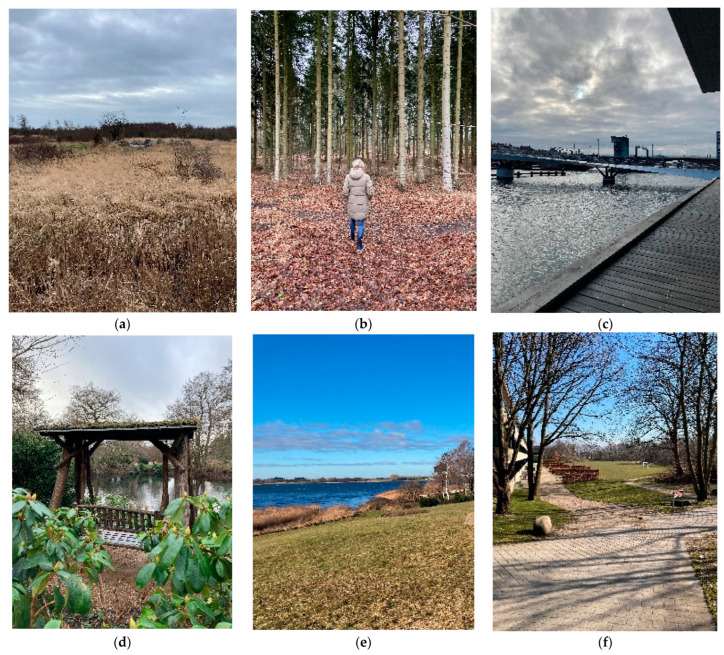
Photo of distinctive greenspace. Note: (**a**) Nature Park at case 1; (**b**) Fir forest at case 2; (**c**) Urban harbor at case 3; (**d**) Campus at case 4; (**e**) Park by the sea at case 5; (**f**) Campus at case 6.

**Figure 3 ijerph-20-06689-f003:**
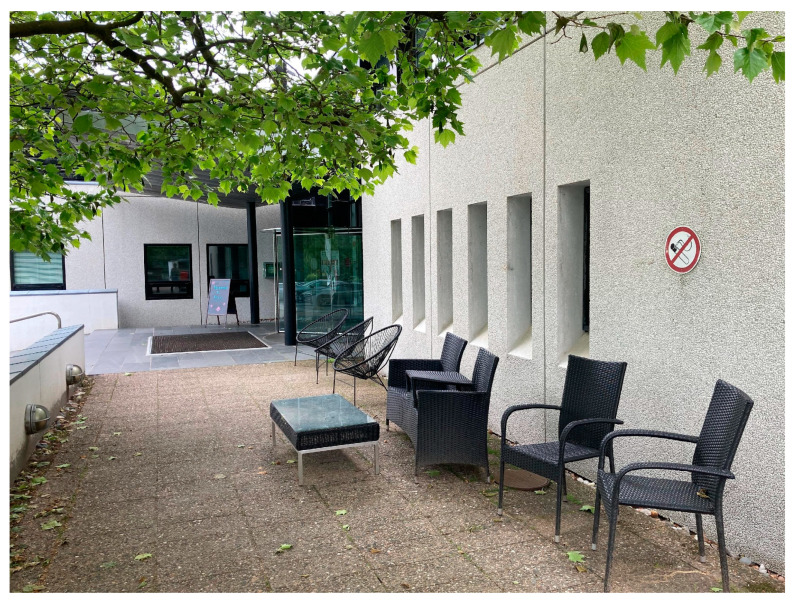
Entrance area at case 4, where the interviewee took mobile phone calls outdoors.

**Figure 4 ijerph-20-06689-f004:**
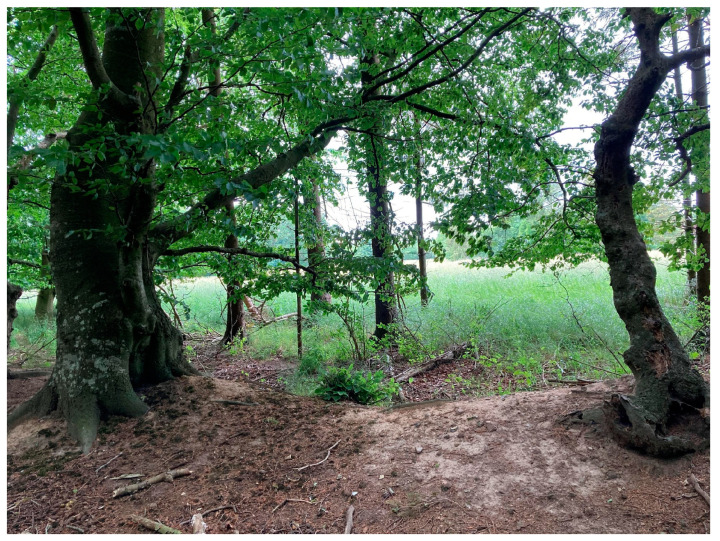
Interviewee 2′s example of a good place to stay. Here, she can sit on a small embankment protected by the forest’s trees while also having a view of a meadow.

**Figure 5 ijerph-20-06689-f005:**
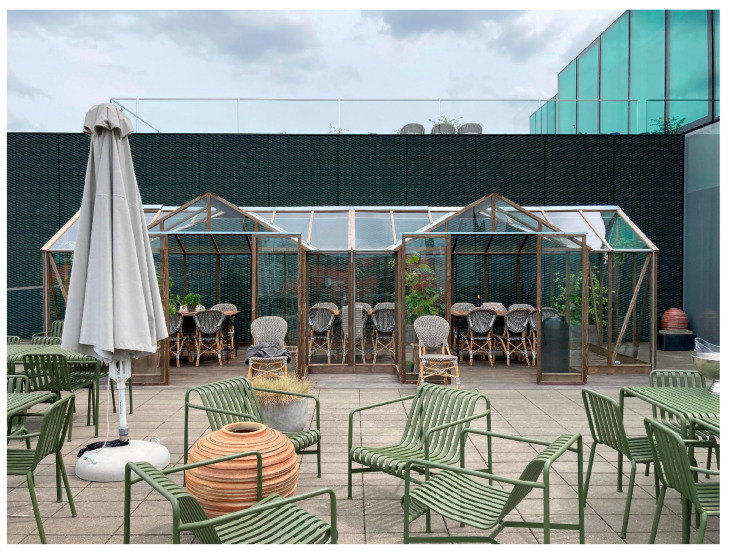
Orangery on the rooftop terrace at case 3 that fulfils several of the interviewees’ wishes for a comfortable outdoor space.

**Figure 6 ijerph-20-06689-f006:**
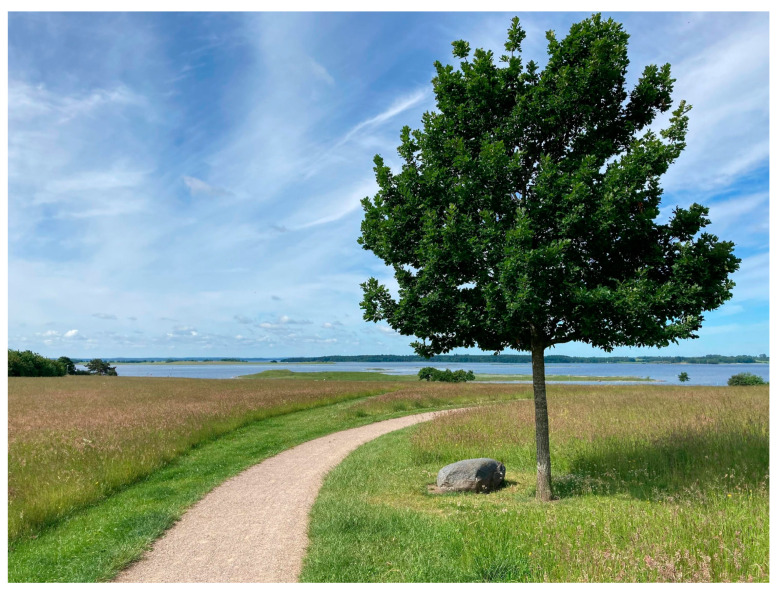
Interviewee 5b chose a route with a magnificent view to get the feeling of air and spaciousness.

**Figure 7 ijerph-20-06689-f007:**
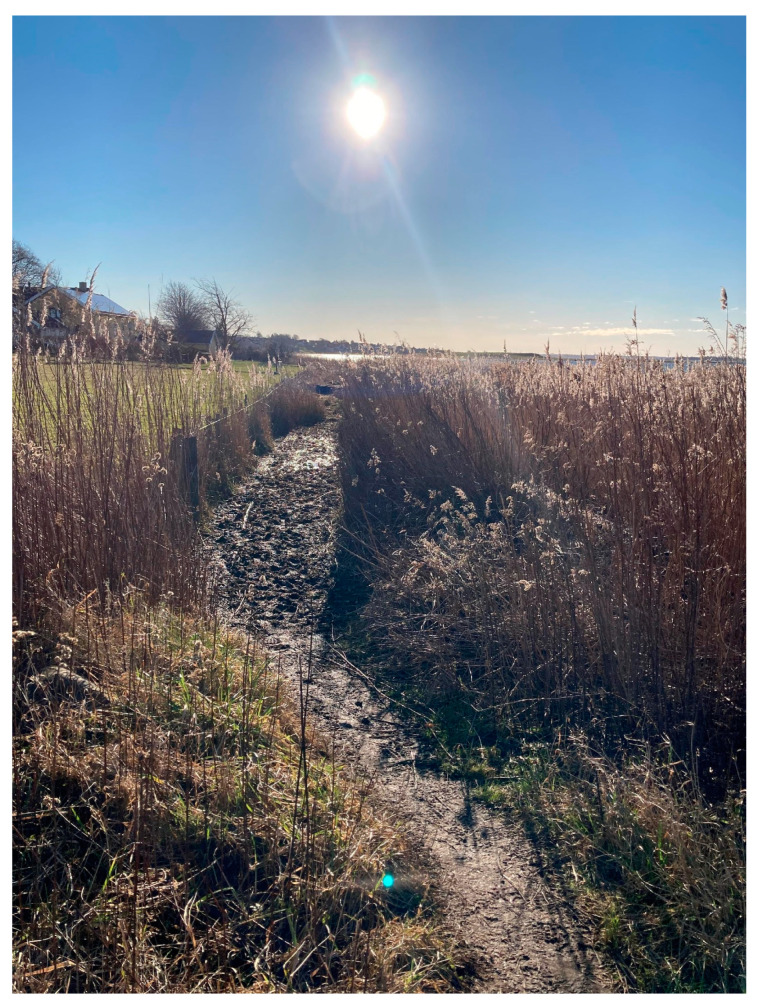
A path on the beach where interviewee 5a sometimes had to walk behind or in front of her colleague instead of next to each other.

**Figure 8 ijerph-20-06689-f008:**
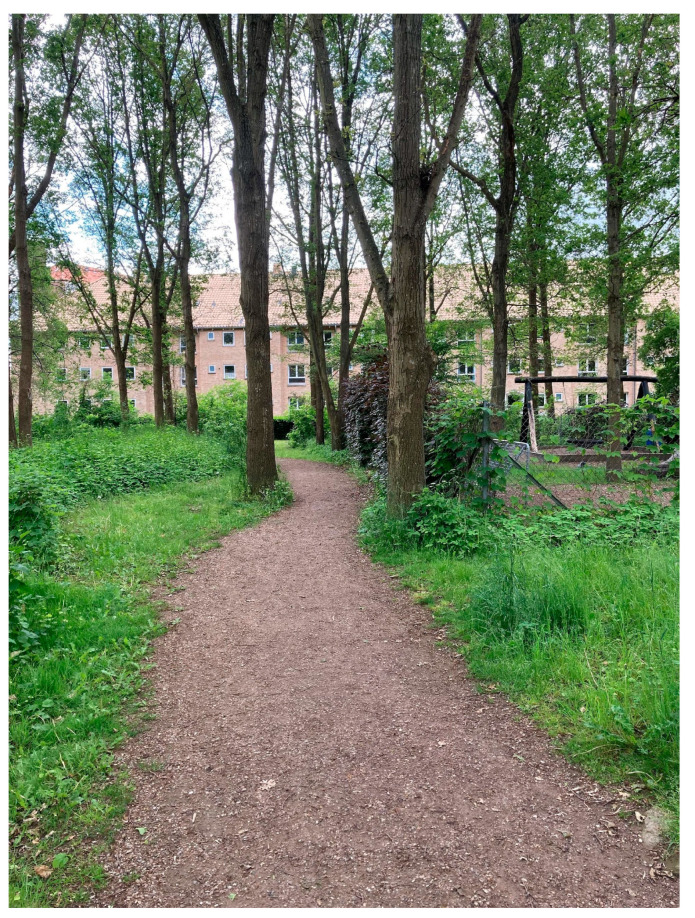
Interviewee 6b’s forest route where she got fresh air, exercise, and work done at the same time. She liked the combination of some natural elements that are kept and some that are grown wild.

**Table 1 ijerph-20-06689-t001:** Overview of cases showing the type of context, company, building, offices, greenspaces, as well as the job function and reference number of the interviewees.

	Case 1	Case 2	Case 3	Case 4	Case 5	Case 6
Context	Periphery of the city	Rural area	City center	Suburban area	Country town	Suburban area
Company	Email marketing agency	Contracting company with specialty in lifts and balconies	Trade association within the field of the built environment	Service company managing a community for deep tech companies	Contracting company	Service company managing a community for deep tech companies
Building type	Block of offices and flats	Office building	Block of offices, flats and museum	Office building	Office building	Office building
Office type	Open-plan office	Open-plan office	Open-plan office	Different types of offices	Open-plan office	Open-plan office
Greenspace	Courtyard, streetscape greenery and nature park	Terraces, lawn, forest and fields	Rooftop terraces, courtyard, urban square, park and urban harbor	Entrance area, terrace and park-like campus	Terraces, garden, streetscape greenery, park and beach	Terraces, campus and wooded area
Interviewee reference number and job function	1, Email marketing specialist	2, Technical designer	3, Consultant	4, Sustainability manager	5a, Manager of working environment 5b, Chief accountant	6a, Customer Service 6b, Head of showings

**Table 2 ijerph-20-06689-t002:** Practical suggestions for promoting outdoor office work with a focus on the physical environment.

Aspects of Key Importance	Promoting Greenspace as a Workplace
Simplicity	Do not overthink! Many work activities can easily be brought outdoors within the existing physical environment. Provide terraces and the like on adjacent outdoor areas, offering wifi and electricity; Map a selection of easily accessed routes of different lengths and clock them; Identify the routes and greenspaces suitable for different types of collaboration, offering possibilities to gather, sit down, and alternate with walk and talks.
Safeness	Identify and map outdoor workspaces and provide facilities/furniture that both offer a view of the surroundings as well as a sense of being sheltered; Be aware of the possible risks and disturbances posed by traffic when mapping the routes for walking and outdoor workspaces; Communicate that working outdoors is both legitimate and recommended by making maps and various equipment clearly visible.
Comfort	Enable shelter from (direct) sun, wind, and rain through sunroofs, parasols, awnings, and the like; Built ‘in-between structures’, such as an orangery, can expand the possibilities for conducting outdoor office work comfortably; Include information about different levels of physical demands of various routes, such as different lengths and topography.
Contact with Nature	Document the characteristics of routes and greenspaces to cater to diverse preferences (sensory experiences, openness, proximity, wildlife, etc.); Identify the greenspaces and routes where you can have a sense of being in contact with nature, away from office (and other) nuisances; Value and develop urban greenspaces.

## Data Availability

The data are not publicly available due to privacy restrictions.

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
