# Peer review of "Greenspace as Workplace: Benefits, Challenges and Essentialities in the Physical Environment"

_ijerph, 2023, doi:10.3390/ijerph20176689_

Round 1

Reviewer 1 Report

Nice article, but without scientific reasoning, I think:
- what is the research method?
- there are 5 companies included in the project, not selected but invited; this means their focus can be altered to fit the project's goals
- are there any objective insights, or just subjective ones?
- what are the goals of the research? just finding that landscape connection is good for our wellbeing?
- conclusions are very generic, there are no strict and fair recommendations for landscape design

Author Response

We are grateful for your valuable comments. Revising the manuscript according to them gave us an opportunity to improve the manuscript.

Questions and answers:

What is the research method? The approach is qualitative using case study research that includes interviews, photos registration, and mapping green spaces.
There are 5 companies included in the project, not selected but invited; this means their focus can be altered to fit the project's goals. Yes, the companies that we selected were in different types of greenspaces. This type of case selection can be called information oriented. We wanted to include cases each with their different qualities to obtain information on a rich variety of greenspace. The participants were invited to take part of the project and participation was voluntary.
Are there any objective insights, or just subjective ones? Results are qualitative and bound to their case. The results may be applied to other contexts to the extent that the descriptions of the cases are taken into consideration with a reasoned judgment about transferability.
What are the goals of the research? just finding that landscape connection is good for our wellbeing? The goal of this research is to obtain knowledge on the benefits and challenges of bringing office work out in different types of greenspaces. We also wanted to identify the factors in the outdoor physical environment that supported office work.  
Conclusions are very generic, there are no strict and fair recommendations for landscape design. This research topic is quite new and unexplored. This is reflected in the results that are of a general and guiding character. More research is needed to get more into depth.    

See also manuscript for text added (highlighted) in response to reviewer comments.

Reviewer 2 Report

Work is adequate, accepted after minor review. Among the aspects that should be improved are:

 - A description of the population and the sample selection criteria are necessary.

- The conducting unstructured individual interviews should be described in greater depth.

- In the second phase, it would be recommendable how the walk and talk are carried out, both in content and in the profile of the interviewer.

-Limitations such as personality, climate or family and professional reconciliation could be included.

Author Response

We are grateful for your valuable comments. Revising the manuscript according to them gave us an opportunity to improve the manuscript. 

Text added in response to your comments:

- A description of the population and the sample selection criteria are necessary.

“The sample selection criteria for the inclusion of companies was pragmatic and primarily based on the individual organization's motivation for participation, a geographical diversification among the participating companies (to strengthen geographical representativeness) and an assessment of the individual company's scope to include a relevant outdoor environment that both created variety between the participating companies and ensured varied experiences.” (p4, 173-178)

“Their age ranged from 22 to 61 and all of them were office workers with a job function that required access to a computer (see Table I). The E-mail marketing specialist, technical designer, customer service employee, manager of working environment and chief accountant were mostly bound by computer work. Whereas the consultant, sustainability manager and head of lettings undertook both computer work and more active work tasks such as showing customers around.” (p5, 216-222)

- The conducting unstructured individual interviews should be described in greater depth.

“The interviews lasted approximately one hour and were held at the companies, starting indoors and ending with a walk in their outdoor environment. Questions concerned how they worked indoors, what options they had for working outdoors and how they used those options. The interviews were also about positive and negative aspects of working outdoors, as well as their preferences and wishes in this regard.” (p5, 227-231)

“The interviewer was the same person that had carried out the research of the first phase of this study – a researcher within the field of landscape architecture with an expertise in connections between human health, nature, and design.” (p5, 241-244)

“Conducting the interview in the outdoor setting gave the interviewer a good understanding of the connections between the experiences talked about and their relation to specific aspects of the physical environment. The photo registration helped capture these connections.” (p6, 252-255)

-Limitations such as personality, climate or family and professional reconciliation could be included.

“In fact, a good share of employees at the companies chose not to participate in the study. This de-selection may be due to different factors such as personal preference for using the outdoors or an incompatible job function with regards to bringing work activities outdoors.” (15, 622-625)

“The topic of this study is in its infancy and needs to be further explored. Transferability may be limited due to various contextual aspects such as type of business, employee characteristics, organisational culture, outdoor environment, and climate [48,52]. Still, the findings of this study may serve as a useful building block for enterprises aiming at integrating the outdoors into everyday working life.” (p15, 630-635)

See also PDF  - manuscript with added text (highlighted in yellow)

Reviewer 3 Report

Dear Authors,

I have had a chance to analyze your paper submitted to the IJERPH. I must say I really enjoyed reading it, both, for its scientific soundness and originality. In the highly-paced world where prime concern seems to be that of employee performance, this paper addresses, simultaneously, the (natural) interest of companies and employee's well-being. 

The methodology of the study, given its exploratory nature, is adequate. The background and discussion are supported by relevant and up to date references.

I believe this research may be of interest to the audience of the journal.

Kind regards,

Good. Final revision required.

Author Response

We are grateful for your valuable comments. Revising the manuscript according to the comments from the 3 reviewers gave us an opportunity to improve the manuscript. See PDF for revisions (highlighted in yellow).
